# Revealing the Beauty Potential of Grape Stems: Harnessing Phenolic Compounds for Cosmetics

**DOI:** 10.3390/ijms241411751

**Published:** 2023-07-21

**Authors:** Mónica Serra, Ana Casas, José António Teixeira, Ana Novo Barros

**Affiliations:** 1Mesosystem, Rua da Igreja Velha 295, São Félix da Marinha, 4410-160 Vila Nova de Gaia, Portugal; ana@mesosystem.com; 2CEB—Centre of Biological Engineering, University of Minho, 4710-057 Braga, Portugal; jateixeira@deb.uminho.pt; 3LABBELS—Associate Laboratory, University of Minho, 4710-057 Braga, Portugal; 4Centre for the Research and Technology of Agro-Environmental and Biological Sciences (CITAB), Institute for Innovation, Capacity Building and Sustainability of Agri-Food Production (Inov4Agro), University of Trás-os-Montes and Alto Douro (UTAD), Quinta de Prados, 5000-801 Vila Real, Portugal

**Keywords:** phenolic compounds, grape stems, cosmetics, antioxidants

## Abstract

Grape stems have emerged as a promising natural ingredient in the cosmetics industry due to their abundance of phenolic compounds, known for their antioxidant and anti-inflammatory properties. These compounds have shown great potential in promoting skin health, fighting signs of aging, and shielding against environmental stressors. With high concentrations of resveratrol, flavonoids, and tannins, grape stems have garnered attention from cosmetic scientists. Research has indicated that phenolic compounds extracted from grape stems possess potent antioxidant abilities, effectively combating free radicals that accelerate aging. Moreover, these compounds have demonstrated the capacity to shield the skin from UV damage, boost collagen production, and enhance skin elasticity. Cosmetic formulations incorporating grape stem extracts have displayed promising results in addressing various skin concerns, including reducing wrinkles, fine lines, and age spots, leading to a more youthful appearance. Additionally, grape stem extracts have exhibited anti-inflammatory properties, soothing irritated skin and diminishing redness. Exploring the potential of grape stem phenolic compounds for cosmetics paves the way for sustainable and natural beauty products. By harnessing the beauty benefits of grape stems, the cosmetics industry can provide effective and eco-friendly solutions for consumers seeking natural alternatives. Ongoing research holds the promise of innovative grape stem-based formulations that could revolutionize the cosmetics market, fully unlocking the potential of these extraordinary botanical treasures.

## 1. Introduction

The winery industry in Portugal and Europe plays a crucial role in the economy, culture, and heritage of these regions, positioning itself as one of the world’s foremost wine-producing areas [1]. However, wine production generates a variety of by-products, including pruning wood, grape pomace, seeds, skins, and stems, among others [2,3,4,5,6,7,8]. Improper management of these by-products can have adverse environmental effects, such as water pollution, soil degradation, and greenhouse gas emissions [9,10]. Recognizing these concerns, the wine industry has increasingly prioritized the adoption of sustainable practices to mitigate its environmental impact and effectively handle the by-products generated during winemaking processes [10,11,12,13,14,15,16,17]. One approach gaining significant momentum is the implementation of circular economy principles, which aim to minimize waste and maximize resource utilization [18].

Scientists have been actively studying these wine residues to unravel their composition and explore potential applications [2]. By discovering novel uses for these by-products, the wine industry can considerably reduce its environmental footprint, contribute to the principles of a circular economy, and potentially unlock new sources of revenue. This represents a pivotal stride towards achieving greater sustainability within the wine industry while minimizing its overall environmental impact.

In recent years, there has been a growing interest in exploring the beauty potential of grape stems in the cosmetics industry [19]. Grape stems, which are typically discarded as waste during winemaking, contain a wealth of valuable phenolic compounds that can offer numerous benefits for skincare and cosmetic products.

Phenolic compounds are known for their antioxidant, anti-inflammatory, and anti-aging properties, making them highly sought-after ingredients in the cosmetic industry. Grape stems are particularly rich in these compounds, including resveratrol, flavonoids, anthocyanins, and tannins. These bioactive compounds have been extensively studied for their ability to protect the skin against oxidative stress, promote collagen synthesis, and improve overall skin health [14].

The use of grape stem extracts in cosmetics holds great promise. These extracts have shown potential in combating skin aging by reducing the appearance of wrinkles, fine lines, and age spots. Their antioxidant properties help neutralize free radicals, which are known to contribute to premature aging and skin damage. Grape stem extracts can also help improve skin elasticity and firmness, giving the skin a more youthful and rejuvenated appearance [20].

Furthermore, the anti-inflammatory properties of grape stem phenolic compounds make them beneficial for soothing and calming irritated skin. They can help alleviate redness, inflammation, and sensitivity, making them suitable for products targeting sensitive or reactive skin types. Additionally, these compounds have shown potential in reducing hyperpigmentation and improving skin tone, providing a more even complexion [16].

Cosmetic formulations incorporating grape stem extracts can range from serums and creams to masks and toners. These extracts can be used as active ingredients or as supportive components to enhance the efficacy of other skincare ingredients. Their inclusion in cosmetic products not only offers potential benefits for the skin but also aligns with the growing demand for natural and sustainable beauty solutions [21].

By harnessing the phenolic compounds present in grape stems, the cosmetics industry can contribute to the reduction of waste generated by the winemaking process while unlocking a new source of valuable ingredients. This utilization of grape stems aligns with the principles of a circular economy, emphasizing the importance of resource optimization and sustainability [22].

In conclusion, the beauty potential of grape stems in cosmetics is a fascinating area of exploration. The phenolic compounds found in these by-products possess a range of beneficial properties for skincare, including antioxidant, anti-inflammatory, and anti-aging effects. Incorporating grape stem extracts into cosmetic formulations can offer consumers natural and sustainable beauty solutions that promote healthy, youthful-looking skin. By recognizing and utilizing the hidden potential of grape stems, the cosmetics industry can contribute to a more eco-friendly and innovative approach to skincare.

## 2. Wine Industry in Portugal and Europe

The wine industry holds significant economic importance in Portugal and Europe, boasting a rich heritage and a diverse array of wine styles and regions. It plays a vital role in the global economy, supporting numerous jobs across various sectors, from grape growers and winemakers to processors, distributors, and retailers. Additionally, the industry serves as a significant cultural and tourist attraction, with vineyards and wineries offering immersive tours and tastings to visitors [1]. 

In Europe, the wine industry stands as a major economic sector, with several countries renowned for producing exceptional wines. According to the OIV [23], European countries dominate global wine production, with France, Italy, Spain, Germany, and Portugal emerging as the top wine-producing nations in Europe, as depicted in Figure 1. Each of these countries boasts its own distinctive wine styles and regions, encompassing a wide spectrum from light and refreshing whites to full-bodied reds and luscious dessert wines [24].

The wine industry holds great historical and economic importance in Portugal, making a significant contribution to the country’s economy. Portugal is renowned for its production of Port wine, a fortified wine exclusively produced in the picturesque Douro Valley region [24,25]. This region’s unique terroir and winemaking traditions have positioned Portugal as a notable player in the global wine market. Moreover, the wine industry in Portugal and Europe at large is deeply intertwined with the regions’ economy, culture, and heritage. The production of wine has shaped the cultural fabric of these regions, with vineyards and wineries often serving as popular tourist destinations. The industry plays a pivotal role in preserving and showcasing the rich heritage associated with winemaking, offering visitors the opportunity to experience the traditions, craftsmanship, and diverse wine styles that define these regions. Additionally, the wine industry contributes significantly to the economies of Portugal and Europe, providing employment across various sectors, from viticulture to production, distribution, and hospitality. The export of wines from these regions further strengthens their economic impact and promotes their unique wine offerings to international markets. In summary, the wine industry in Portugal and Europe serves as a cornerstone of their economies, cultures, and legacies. The production of world-renowned wines, such as Port wine in Portugal, adds to the allure of these regions and continues to attract wine enthusiasts and connoisseurs from around the globe.

### 2.1. By-Products from the Winery Industry and Their Valorization

The wine production process generates a wide range of by-products, with Europe alone producing an estimated 14.5 million tons [2,3,9]. To gain a comprehensive understanding of these by-products, it is necessary to examine the various stages of winemaking. Vines are pruned annually to optimize grape production and encourage vine growth, resulting in significant quantities of pruning wood as a major by-product. Pruning wood can be utilized directly as organic material in vineyards or undergo composting processes. It also has potential applications in the production of wood pellets [4,9].

The winemaking process begins with the harvesting of grapes, which are then transported to the winery for crushing and pressing. During this stage, the grape berries are broken to combine the juice, pulp, seeds, skins, and stems, followed by the separation of the juice from the solid components. Subsequent stages, such as fermentation and clarification, convert the grape’s sugar into alcohol and remove any suspended insoluble matter from the wine. As a result, by-products, such as grape pomace and stems, are generated at each step of the process, as depicted in Figure 2A [2,3,4,5,6,7,8].

The generation of by-products during winemaking is an inherent outcome that cannot be avoided. While these by-products themselves are not toxic, their high organic matter content can pose environmental challenges if not managed properly. Inadequate management of these by-products can contribute to issues such as water pollution, soil degradation, and greenhouse gas emissions [9,10]. Recognizing the environmental impact associated with the substantial production of winemaking by-products, the wine industry has increasingly focused on adopting sustainable practices to mitigate its environmental footprint and effectively manage these by-products [10,17].

By embracing the principles of the circular economy [18], the wine industry aims to reduce waste generation and explore innovative ways to utilize by-products such as pruning wood, seeds, skins, stems, and lees. This entails implementing strategies to minimize waste, optimize the management of by-products, and discover new avenues for their utilization. By finding novel uses for these by-products, the wine industry can contribute to a more sustainable and circular approach, where materials are kept in use for as long as possible, reducing the strain on the environment.

The circular economy model encourages minimizing waste and maximizing resource utilization, as illustrated in Figure 2B. Within the context of winemaking, this involves implementing strategies to reduce waste generation, optimize by-product management, and explore new avenues for their utilization. By discovering innovative applications for these by-products, the wine industry can contribute to a more sustainable and circular approach, where materials are utilized to their fullest extent, thus reducing the environmental impact. Efforts to develop sustainable practices and adopt circular economy principles in winemaking are crucial for reducing the industry’s environmental footprint and fostering a more sustainable future. By valuing and effectively utilizing by-products, the wine industry can move towards a more environmentally conscious and resource-efficient approach to winemaking.

In line with the principles of the circular economy, scientists have extensively studied the composition and potential applications of winemaking residues [14,26]. The main by-products of the wine industry include pruning wood and grape marc (grape pomace), which comprises stalks (stems), skins, and seeds [5,7,17,27,28]. The composition of these residues can be influenced by several factors such as climate, soil type, fertilization, grape variety, maturation, and harvest time [5,29,30]. These residues exhibit complex compositions, encompassing polysaccharides, proteins, fibers, minerals, lignin, cellulose, hemicelluloses, vitamins, amino acids, and phenolic compounds. Each residue from the winemaking process possesses a distinct and characteristic composition [8,17,28,30,31,32]. For instance, pruning wood is primarily composed of lignin, cellulose, hemicellulose, and minerals like potassium and calcium. Due to its composition, it can be bio converted into raw materials for cultivating microorganisms that yield value-added products such as biodiesel, alcohol, and lactic acid [6,13,30].

Grape pomace, on the other hand, is recognized for its high content of phenolic compounds, including phenolic acids, flavonoids, tannins, and stilbenes [8,17,33,34,35,36]. These compounds are widely distributed in the plant kingdom and possess various significant biological and pharmacological properties, such as anti-inflammatory and antioxidant activities, cardioprotective and neuroprotective potentials, and anticancer and antibacterial properties [5,9,27,30,31,35,36,37,38,39]. The in-depth understanding of the composition of these winemaking residues opens avenues for their potential utilization in various applications. By harnessing the valuable components present in these residues, it becomes possible to extract and develop novel products with beneficial properties, contributing to the sustainable and circular utilization of winemaking by-products.

### 2.2. Grape Stems: Chemical and Phenolic Composition and Their Biological Potential

Grape stems, also referred to as grape stalks, are the sturdy branches that extend from the vine trunk and provide support to the grape bunch or cluster [5,40]. Playing a vital role in the vine’s structure, these stems help in distributing water, nutrients, and hormones to all the grape berries [40]. The structural characteristics of grape stems are illustrated in Figure 3. During the winemaking process, grape stems are typically separated from the grapes before fermentation commences. This practice is carried out due to their composition, which can introduce an unpleasant bitterness to the wine if they remain in contact with the juice during fermentation [9,40,41,42,43]. In fact, grape stems account for up to 14% of the total weight of solid wastes generated during wine production [28].

However, it is worth noting that grape stems possess their own unique chemical and phenolic composition, which has drawn scientific interest. While their presence in wine is undesired, grape stems exhibit potential biological properties that make them valuable for alternative applications.

Indeed, the composition of grape stems is influenced by various factors, including grape variety, maturation state, size, and extraction methods, resulting in variations in their biological characteristics [30,32,40,44]. Chemically, grape stems predominantly consist of lignocellulosic materials, such as lignin, cellulose, hemicelluloses, and ash [32,40,41,45]. They also contain significant amounts of monosaccharides, particularly glucose and xylose, as well as smaller quantities of mannose, arabinose, and galactose [44,46,47]. Furthermore, grape stem extracts have been found to exhibit a high content of essential minerals like sodium (Na), magnesium (Mg), calcium (Ca), and potassium (K) [48].

Regarding their phenolic composition, grape stems contain bioactive phenolic compounds that are widely distributed in the plant kingdom. These compounds are produced by plants as a response to environmental aggressions such as UV solar radiation, biotic and abiotic stress, and pathogens [2,41]. 

Phenolic compounds generally possess an aromatic ring structure with at least one hydroxyl group. They exhibit a wide range of chemical structures, from simple molecules, like phenolic acids, to more complex polymeric structures, like tannins. Various classes of phenolic compounds exist, including flavonoids (flavanols, flavonols, flavones, anthocyanins), phenolic acids (hydroxybenzoic acids and hydroxycinnamic acids), stilbenes, and tannins, each with its distinct chemical structure and biological activity [2,27,49]. Table 1 provides some examples of these phenolic compounds.

The biological potential of phenolic compounds extracted from grape stems has been extensively studied, revealing various beneficial activities, including antioxidant, anti-inflammatory, anti-aging, anticancer, and antimicrobial effects [27,31,40,41,50]. 

Researchers have conducted in-depth analyses of grape stem extracts to identify their phenolic compound composition and explore their biological potential. For instance, Makris et al. [51] identified several phenolic compounds in grape stem extract, including gallic acid (a hydroxybenzoic acid), trans-resveratrol, ε-(or α-)viniferin (stilbenes), flavonols such as quercetin-3-*O*-glucuronide and quercetin-3-*O*-rutinoside (rutin), and the flavanonol astilbin. Similarly, Nieto et al. [52] found that optimal grape stem extracts are rich in flavonoids, particularly catechin, epicatechin, and procyanidin B1 from the flavan-3-ols subclass. They also observed the presence of quercetin-3-*O*-glucuronide from the flavonols subclass, as well as significant amounts of phenolic acids, including gallic acid, vanillic acid, syringic acid (hydroxybenzoic acids), and *trans*-caftaric acid (a hydroxycinnamic acid). The extract also contained substantial levels of stilbenes, such as ε-viniferin, *trans*-resveratrol, and trans-resveratrol tetramer, along with lower concentrations of anthocyanins, namely malvidin-3-*O*-glucoside. Jara-Palacios et al. [53] conducted a similar study on grape stem composition and reported comparable findings. They identified flavan-3-ols, including catechin, epicatechin, procyanidin B1, procyanidin B4, and procyanidin B2 3-*O*-gallate, as the main phenolic compounds in the extract. Flavonols, like quercetin 3-O-glucuronide and quercetin 3-*O*-glucoside, as well as phenolic acids, such as gallic acid and caftaric acid, were also present. In addition, the exploration of stilbenoids in grape stem extracts using ultrasound-assisted extraction revealed the presence of other stilbenes beyond resveratrol and viniferin. These included piceatannol, isorhapontigenin, and vitisin-B [54]. These studies collectively demonstrate the diverse range of phenolic compounds present in grape stem extracts and their potential biological activities, further highlighting the value of grape stems as a source of bioactive compounds. Phenolic compounds in grape stem extracts are well known for their strong antioxidant properties, which have been extensively studied by researchers. R. Domínguez-Perles et al. [55] specifically investigated the phenolic content of grape stem extracts from the red variety Touriga Nacional. They determined the total phenolic content to be 64.25 mg GAE/g dw, total ortho-diphenols at 42.61 mg GAE/g dw, total flavonoids at 71.96 mg CE/g dw, and total anthocyanin at 1.99 mg/g dw, after optimizing the extraction conditions. The researchers also studied the antioxidant activity and observed a higher scavenging capacity by the extracted phenolic compounds. Other researchers who explored extraction conditions discovered extracts that were rich in phenolic compounds, with the most abundant ones being catechin (flavan-3-ol), gallic acid (hydroxybenzoic acid), a derivative of quercetin (flavonol), and stilbenes (resveratrol and viniferin). Additionally, Jiménez-Moreno et al. [56] related the phenolic composition of the extract to its antioxidant potential and concluded that stilbenes do not contribute significantly to the antioxidant properties of grape stem extracts. Instead, malvidin-3-glucoside and quercetin were identified as the main compounds responsible for the antioxidant capacity of the extracts.

Numerous investigations have indicated the antioxidant effects of phenolic compounds from grape stems. As a result, scientists have explored this property in endothelial and muscle cells. They analyzed the composition of extracts from several grape varieties and found a similar phenolic composition among all extracts, although the content of each isolated compound varied. The difference in antioxidant activity was attributed to the varying amounts of each isolated phenolic compound. The Mandilaria variety exhibited a better antioxidant effect on endothelial and muscle cells, which was attributed to the higher amounts of specific phenolic compounds (such as trans-resveratrol, gallic acid, quercetin, catechin, epicatechin, and rutin) present in this variety compared to others [57]. Anastasiadi et al. [58] and Spatafora et al. [59] also explored the extraction of phenolic compounds from grape stems and found that these extracts are rich in natural antioxidant compounds. 

Phenolic compounds are also recognized for their potent antimicrobial activity, which makes them useful in various applications for controlling microbial growth, including bacteria. To evaluate the phenolic profile and assess the antioxidant and antimicrobial properties, Silva et al. [60] characterized extracts from grape stems of two varieties, Touriga Nacional and Preto Martinho. They determined a total phenolic content of 45.9 mg/g and 226.8 mg/g, and a total tannin content of 7.76 mg/g and 22.15 mg/g, respectively. Flavan-3-ols were identified as the main phenolic compounds in the extracts, with catechin, epicatechin, and gallocatechin gallate being expressed. Phenolic acids, including p-coumaric acid and ferulic acid (hydroxycinnamic acids), as well as vanillic acid, gallic acid, and protocatechic acid (hydroxybenzoic acids), were also highly present. Trans-resveratrol was identified as a stilbene compound. In terms of biological activity, the results demonstrated a good antioxidant capacity, and microbiologically, the extracts were effective in inhibiting gram-positive bacteria.

Grape stem extracts have been found to exhibit antimicrobial activity [50]. Dias et al. [61] conducted a study to identify the phenolic compounds responsible for this activity and evaluated the antimicrobial activity of extracts against digestive pathogens. They discovered that extracts from the Fernão Pires variety were more effective against Listeria monocytogenes and Pseudomonas aeruginosa, while extracts from the Tinta Amarela variety showed stronger activity against Pseudomonas aeruginosa. The phenolic compounds responsible for the antimicrobial activity were identified as kaempferol-3-*O*-rutinoside (flavonol), kaempferol-3-*O*-glucoside (flavonol), and caftaric acid (hydroxycinnamic acid).

Due to the demonstrated biological activities, grape stem extracts have potential applications in the cosmetic and pharmaceutical industries. Leal et al. [62] analyzed the composition of extracts from various grape varieties and identified catechin (flavan-3-ol), caftaric acid (hydroxycinnamic acid), and quercetin-3-*O*-rutinoside (flavonol) as the major phenolic compounds. Notably, all studied varieties exhibited a significant content of anthocyanidins, particularly malvidin-3-*O*-galactoside and malvidin-3-*O*-glucoside, in the grape stem extracts of the Tinta Roriz variety. The authors also investigated the antioxidant, antimicrobial, anti-inflammatory, and anti-aging activities of grape stem extracts. Grape stem extracts demonstrated good antioxidant potential with effective scavenging capacity, consistent with previous studies [63]. In terms of antimicrobial activity, the extracts showed efficacy in inhibiting Gram-positive bacteria in the gastrointestinal segment and diabetic foot ulcers, particularly Enterococcus faecalis and Staphylococcus aureus [63].

The anti-inflammatory activity of grape stem extracts was assessed by measuring the inhibition of nitric oxide (NO) production. All the studied extracts exhibited a strong ability to inhibit NO production by macrophages, with the extracts from the Arinto variety being the most effective [62].

The overexpression of tyrosinase and elastase enzymes is associated with skin aging. Grape stem extracts were found to possess anti-elastase and antityrosinase activities, suggesting their potential as anti-aging agents. Among the extracts, those from the Syrah variety showed the highest effectiveness in inhibiting the activity of tyrosinase and elastase, indicating their possible application in the cosmetic industry [62]. 

Some phenolic compounds found in grape stem extracts have shown potential in preventing or inhibiting the growth of cancer cells, demonstrating anticancer activity. Quero et al. [64] studied the phenolic composition of grape stems and evaluated the activity of stem extracts on different breast cancer cell lines. They found that catechin, quercetin, ε-viniferin, and *trans*-resveratrol were the most abundant phenolic compounds in the extract. The extract was found to decrease the growth of breast cancer cells and exhibit antioxidant effects on intestinal barrier cells. Sahpazidou et al. [65] also explored the phenolic composition of stem extracts from various grape varieties and investigated their inhibitory effects on the growth of human cancer cell types, such as breast, kidney, thyroid, and colon. The extracts exhibited inhibition of cell growth in different types of cancer, particularly renal and thyroid cancer cells. These findings suggested the potential benefits of grape extracts in preventing and treating various types of cancer.

Furthermore, individual phenolic compounds present in grape stem extracts have been studied for their specific biological activities. Resveratrol, for example, has been suggested to have neuroprotective effects and could be beneficial in the treatment of neurological disorders such as Alzheimer’s disease [66]. Catechins and epicatechins, which are also found in grape stem extracts, have been investigated for their potential neuroprotective effects [67]. In summary, grape stem extracts are rich in phenolic compounds, particularly flavan-3-ols, such as catechin. Anthocyanins are present in relatively low amounts in these extracts. The composition of phenolic compounds in the extracts can vary depending on factors such as grape variety, cultivation region, soil composition, UV radiation, and altitude [29,68,69,70,71,72]. Grape stem extracts have demonstrated various biological activities, including antioxidant, antimicrobial, anti-inflammatory, and anti-aging properties. They have shown efficacy against Gram-positive bacteria and have exhibited inhibitory effects on cancer cell growth. The identified phenolic compounds, such as kaempferol-3-O-rutinoside, kaempferol-3-O-glucoside, and caftaric acid, have been associated with the antimicrobial activity of grape stem extracts. These findings suggest the potential application of grape stem extracts in the cosmetic and pharmaceutical industries. However, further research is needed to explore the specific mechanisms of action and potential therapeutic applications of grape stem extracts and their phenolic compounds.

### 2.3. Extraction Procedures for Phenolic Compounds

The choice of extraction method is crucial for the recovery of bioactive compounds from grape stems and other plant sources. Several extraction methods are available, each with its advantages and disadvantages [7,30,37,41,73]. Solvent extraction is a commonly used method, where a suitable solvent is used to extract compounds from the plant material. However, this technique has drawbacks such as long extraction times, high labor requirements, and the use of large quantities of solvents, which can be costly and have environmental impacts [37,52]. In general, the extraction process involves several steps (as shown in Figure 4).

Researchers have explored different conditions and solvents to improve the extraction efficiency of phenolic compounds from grape stems. For instance, optimal conditions using a solid–liquid extraction with ethanol–water mixture, temperature, and time were established to extract antioxidant compounds [55]. Temperature and solvent selection were also found to be crucial parameters in solvent extraction, with ethanol at 60 °C yielding better results [74]. Additionally, the concentration of ethanol in the solvent was found to be a determinant variable for extracting specific compounds such as gallic acid, catechin, and trans-resveratrol [56]. Acetone, ethanol, and ethyl acetate were also investigated as solvents for extracting different phenolic compounds, revealing that each solvent exhibited better efficiency for specific compounds [75].

To enhance the extraction process, researchers have explored pretreatment techniques such as instantaneous controlled pressure drop (DIC) and ultrasound-assisted extraction (UAE). DIC pretreatment, in combination with conventional solvent extraction, improves the extraction of phenolic compounds by rupturing plant cell walls, resulting in increased extraction efficiency [76]. UAE utilizes ultrasound waves to break down cell walls, enhancing extraction efficiency. This technique has advantages such as low instrumental requirements, cost-effectiveness, and shorter extraction times [60].

Other extraction techniques such as pressurized liquid extraction (PLE) and superheated liquid extraction (SHLE) have also been studied. PLE utilizes high pressure and temperature to extract bioactive compounds, with advantages including high extraction efficiency and low solvent consumption. However, optimal extraction temperatures need to be determined [37,77]. SHLE, a similar technique to PLE, has been shown to yield higher antioxidant content when using superheated ethanol as the solvent [78]. New extraction methods like high voltage electric discharge (HVED) have been explored. HVED disrupts cell walls through electrical discharges, improving the extraction of target compounds. This method has shown success in extracting flavan-3-ols and flavonols [79].

Combining different techniques has also been investigated. For example, ultrasound-assisted extraction combined with pulsed electric field (PEF) pretreatment has demonstrated an increase in the yield of phenolic compounds [80].

Overall, the selection of an appropriate extraction method is crucial for the recovery of bioactive compounds from grape stems. Researchers have explored several conditions, solvents, and pretreatment techniques to improve the efficiency and yield of phenolic compound extraction. Continued research and development in extraction methods are vital for enhancing the extraction process and obtaining higher yields of bioactive compounds from grape stems.

## 3. Cosmetic Industry

The cosmetic industry is a thriving sector with a significant impact on the global economy, offering a wide array of products that aim to enhance and beautify the skin, hair, and overall appearance of individuals. It encompasses various product categories, including makeup, skincare, hair care, fragrances, and other personal care items [81,82,83].

The European cosmetic industry holds a prominent position in the global market, renowned for its size and innovative contributions. According to the European Cosmetics Association (Cosmetics Europe), the European cosmetics and personal care market reached a substantial value of EUR 78.6 billion in 2019. This industry is a significant contributor to the European Union’s economy, providing employment opportunities to over 1.7 million people and exporting products to countries worldwide.

The cosmetic industry is characterized by intense competition, driving companies to continuously seek ways to gain a competitive edge. This pursuit involves offering unique products, ensuring superior quality, and maintaining affordability. Within the European Union, cosmetic products are regulated by a comprehensive framework that sets rigorous safety standards for both products and ingredients. This regulatory framework guarantees consumer protection and reinforces the industry’s commitment to safety and compliance [82,84].

An emerging trend in the cosmetic industry is the growing demand for natural and organic products. Consumers are increasingly mindful of the ingredients used in their cosmetic products and are seeking options that are free from harmful chemicals while being environmentally friendly. In response to this demand, many companies have started offering natural and organic cosmetic products that avoid synthetic fragrances, preservatives, and other potentially harmful chemicals. This trend reflects a heightened consumer awareness of the impact of personal care products on their well-being and the environment [81,85,86,87,88,89,90].

In conclusion, the cosmetic industry is a rapidly expanding and highly competitive sector that caters to individuals’ personal grooming and hygiene needs. It is fueled by innovation, effective marketing strategies, and adherence to regulatory compliance. The industry is evolving to meet the growing demand for natural and organic products as consumers prioritize safety, sustainability, and environmental consciousness in their purchasing decisions.

### 3.1. Cosmetics

Cosmetics, as defined in Regulation (EC) No. 1223/2009 of the European Parliament and of the Council dated 30 November 2009 [91], encompass a wide range of products intended for personal care, hygiene, and beautification purposes. This definition includes substances or mixtures that are applied externally to various parts of the body, such as the skin, hair, nails, lips, external genital organs, teeth, and mucous membranes of the oral cavity [84]. The primary goal of cosmetics is to enhance the appearance, texture, and overall health of the skin, hair, and nails. These products are utilized for cleansing, moisturizing, protecting, and accentuating the natural beauty of these tissues [81]. Furthermore, cosmetics can effectively address pigmentation concerns, such as hyperpigmentation (darkening of the skin) and hypopigmentation (lightening of the skin). These concerns can arise due to multiple factors, including sun exposure, hormonal changes, and aging [81,92]. Moreover, numerous cosmetic products are specifically formulated to minimize the appearance of fine lines and wrinkles, improve skin texture, and promote a more youthful look. They can also be utilized to treat and prevent acne as well as provide protection against harmful UV rays from the sun. This preventive measure aids in the prevention of sunburn, skin damage, and the reduction of the risk of skin cancer [81,93].

#### 3.1.1. EU Cosmetic Regulation 

Cosmetics in Europe are subject to regulation by the European Union (EU) under the Cosmetics Regulation (EC) No 1223/2009 [91]. This regulatory framework governs the production and marketing of cosmetic products within the EU, aiming to harmonize standards and ensure the safety of these products for consumers. To ensure consistent quality and adherence to standards, Good Manufacturing Practices (GMP) guidelines play a crucial role. Guidelines such as ISO 22716 (Cosmetics—Good Manufacturing Practices (GMP)—Guidelines on Good Manufacturing Practices) and ISO 16128 (Cosmetics—Guidelines on technical definitions and criteria for natural and organic cosmetic ingredients) provide valuable tools for maintaining quality standards during production [94]. According to EU regulation [94,95], cosmetic products must meet certain requirements. Safety and quality are of utmost importance. A cosmetic product must be safe for human health when used under normal or reasonably foreseeable conditions. It should possess the appropriate quality for its intended use and must not pose any risks to human health or the environment. Therefore, the product must undergo evaluation for potential toxicity, irritation, and sensitization [94]. The efficacy of the cosmetic product is another important requirement. It should effectively fulfill its intended function, such as moisturizing or providing anti-aging effects. Additionally, the product’s stability and microbiological quality must be evaluated [96]. This involves assessing the product’s ability to maintain stability over time, including resistance to temperature and light. Furthermore, the presence of microorganisms in the product is evaluated to ensure both safety and stability [94,95].

#### 3.1.2. The Skin and the Effect of Phenolic Compounds in the Skin

The skin, the largest organ in the human body, covers an average area of 2 square meters in adults [97]. It is a complex organ consisting of multiple layers, each serving a specific function. The skin acts as a protective barrier against various external threats, including physical, chemical, and biological factors. Additionally, it plays a vital role in regulating body temperature, maintaining hydration, and facilitating sensory perception [98].

Generally, the skin is structured into three main layers: the epidermis, dermis, and subcutaneous tissue.

The epidermis, the outermost layer of the skin, is primarily composed of keratinocytes that produce keratin, a protein that provides strength and protection against external factors such as UV radiation and pollutants. The epidermis is divided into several layers, with the outermost layer known as the stratum corneum. The stratum corneum consists of dead skin cells and acts as a protective barrier, preventing water loss and shielding the underlying skin layers from damage. It is impermeable to most molecules, including active substances [99]. As we age, the turnover of skin cells slows down, resulting in a thinner stratum corneum and increased water loss, which can contribute to dry and flaky skin [81,100]. Beneath the epidermis lies the dermis, which mainly consists of fibroblasts responsible for synthesizing collagen, elastin, glycosaminoglycans, and other proteins that provide structure and elasticity to the skin. With age, the synthesis of these components can decline, leading to the formation of wrinkles and sagging skin [81,100,101].

The subcutaneous tissue, the deepest layer of the skin, regulates body temperature and stores fat. As we age, the subcutaneous tissue becomes thinner, resulting in a loss of volume and the appearance of sagging skin [81,100,102].

In addition to the natural aging process, factors such as sun exposure, pollution, and smoking can contribute to premature skin aging. These factors can cause the breakdown of collagen and elastin fibers and the production of free radicals, which can damage skin cells and accelerate the aging process [93,103].

Phenolic compounds have demonstrated a wide range of effects on the skin, with their specific targets depending on the compound and the targeted area of the skin [22,85,89,104,105,106]. They can target various structures within the skin, including keratinocytes, fibroblasts, and the extracellular matrix. Some polyphenols have also shown the ability to inhibit melanin production in the epidermis, helping to reduce hyperpigmentation and improve skin tone. One of the significant mechanisms of action for phenolic compounds is their ability to scavenge free radicals and reduce oxidative stress. Oxidative stress is a major contributor to skin aging, and phenolic compounds can help protect the skin from damage caused by reactive oxygen species (ROS) and other free radicals (Figure 5).

Furthermore, polyphenols can exhibit anti-inflammatory effects on the skin, offering protection against damage caused by environmental factors like UV radiation and pollution. Therefore, researchers have investigated the effects of cosmetic formulations based on phenolic compounds on human skin. The Cosmetic Ingredient Review Expert Panel (CIR), an independent scientific body, has reviewed and assessed the safety of 24 grape-derived ingredients used in cosmetics. The CIR determined that these ingredients are safe for use in cosmetics at the current practices of use and concentration, serving as skin-conditioning agents, antioxidants, flavoring agents, and/or colorants [107].

#### 3.1.3. Evaluation of Safety, Quality, and Efficacy of Cosmetic Formulations Containing Phenolic Compounds 

Numerous scientific studies have explored the efficacy and safety of cosmetic formulations containing phenolic compounds. These studies have investigated various types of formulations, including creams, lotions, serums, and toothpaste [104]. Wagemaker et al. [108] conducted a study to evaluate the influence of a topical antioxidant formulation based on green tea polyphenol (GTP) on the inflammatory response of Japanese skin. The results demonstrated that the formulation effectively penetrated the skin, promoting the recovery of the skin barrier and inhibiting the release of inflammatory cytokines. The authors concluded that the GTP-based formulation is an effective antioxidant that can delay the signs of skin aging. Maluf et al. [109] assessed the efficacy and safety of grape pomace extract as an antioxidant raw material for cosmetic formulations. The study evaluated the extract’s antioxidant effect, cytotoxicity, cytoprotection, and cell morphology. The results showed that the grape marc extract exhibited significant radical scavenging activity and did not induce cytotoxicity or morphological changes in 3T3 fibroblast cells. The authors concluded that grape marc extract could be a safe and effective raw material for antioxidant cosmetics.

Taofiq et al. [110] investigated the cell viability and cytotoxic effects of mushroom ethanolic extracts on keratinocytes and fibroblasts. The results indicated that the extracts were not toxic to the cells, suggesting their safe use as crude extracts in cosmetic formulations.

Rafique et al. [111] studied the anti-aging potential of a cream containing grape seed extract. They measured various skin parameters, including moisture level, pH value, sebum content, pore size, elasticity, and roughness. The results showed increased skin hydration, improved elasticity, and reduced pore size and roughness after topical application of the cosmetic. The authors concluded that the cream containing grape seed extract could be used as a natural skin lightener, moisturizer, and potential anti-aging agent. The formulation also exhibited potential for reducing skin redness and acne.

Sharif et al. [112] investigated the effects of an emulsion containing 2% M. hamburg grape seed extract on the cheek skin of human males. The study utilized non-invasive tests and found that the emulsion reduced skin melanin content, sebum content, and erythema while increasing skin elasticity over an 8-week period. The stable formulation was considered a potential natural skin lightener, moisturizer, and anti-aging agent.

Waqas et al. [113] developed a cosmetic formulation containing grape seed extract and assessed its effects on various skin parameters over 12 weeks. The results showed a gradual decrease in skin melanin, increased skin moisture content, improved skin elasticity, and diminished roughness, scaliness, and wrinkles. The formulation was considered cost-effective and safe for use as an anti-aging cosmetic.

In addition to these studies, phenolic compounds have been found to inhibit collagenase and elastase, enzymes responsible for the breakdown of collagen and elastin in the skin. This inhibition helps prevent the development of wrinkles, sagging skin, and other signs of aging [114]. Some phenolic compounds also possess the ability to absorb UV and blue light radiation, acting as physical filters to protect the skin. However, it has been observed that the protection provided by grape seed extracts alone may not be sufficient for effective sun protection. Combining the extracts with sun protection formulations has shown improvements in protection against UV radiation and blue light [115]. Furthermore, grape seed extract has been shown to increase the viability and proliferation of human dermal fibroblasts while protecting the cells from damage caused by UVA rays [116].

Hubner et al. [117] evaluated the safety and clinical efficacy of a sunscreen formulation containing grape pomace extract as a bioactive ingredient. They tested various emulsions with different concentrations of the extract and UV filters and found that a formulation with a high concentration of extract (10% *w*/*w*) and UV filters demonstrated the highest SPF value (16). This formulation was clinically evaluated and compared to a formulation without the extract. The grape pomace extract formulation proved to be more photostable and offered better protection against UVA radiation. In safety tests, both formulations did not induce any skin irritability, sensitization, phototoxicity, or photosensitization. However, the formulation with grape pomace extract exhibited greater clinical photoprotection, requiring more time to produce an erythematous response on the skin compared to the formulation without the extract.

To evaluate the efficacy of cosmetic formulations containing phenolic compounds, researchers have conducted in vitro and in vivo studies. In vitro studies provide valuable information on the biological activity of the compounds, while in vivo studies help determine their effectiveness and safety in human subjects. The mentioned studies demonstrate the potential of phenolic compounds, such as green tea polyphenol, grape pomace extract, and grape seed extract, in cosmetic formulations for antioxidant effects, anti-aging properties, skin barrier recovery, inhibition of inflammatory cytokines, increased skin hydration, improved elasticity, and protection against UV radiation. These findings contribute to the understanding of the benefits and safety of using phenolic compounds in cosmetic products.

#### 3.1.4. Evaluation of Penetration of Phenolic Compounds in the Human Skin 

Penetration of active substances into the skin is a complex process involving multiple steps. The first step involves the diffusion of the active substance within the topical formulation, which is influenced by the substance’s physicochemical properties and the formulation itself. The second step is the release of the active substance from the formulation, which can be affected by factors like viscosity, pH, and temperature. The most challenging step is the actual penetration of the active substance through the skin, given its heterogeneous nature [97].

Studies have investigated the skin permeation of phenolic compounds. Marti-Mestres et al. [118] examined the percutaneous penetration of three natural antioxidant compounds (caffeic acid, chlorogenic acid, and oraposide) into excised pig skin in vitro. Their findings demonstrated that caffeic acid and chlorogenic acid could reach all layers of the skin, while oraposide was primarily detected in the outermost layer, the epidermis.

The penetration of phenolic compounds into deeper skin layers may depend on the delivery system utilized. Butkeviciute et al. [119] explored the release and penetration of phenolic compounds (from apple extract) from various formulations, such as emulsion, emulgel, gel, ointment, and oleogel, into human skin layers. They also evaluated the antioxidant potential of the compounds in the skin. Initially, they assessed the penetration of isolated phenolic compounds and found that only three compounds (chlorogenic acid, rutin, and catechin) could penetrate the skin layers. Chlorogenic acid exhibited the highest penetration into the dermal and epidermal layers. The researchers then studied the penetration of phenolic compounds incorporated into cosmetic formulations and discovered that only chlorogenic acid, rutin, and catechin were released from the formulations. Chlorogenic acid and quercetin, incorporated into all cosmetic formulations, successfully penetrated the dermis and epidermis. Rutin, on the other hand, could only penetrate both skin layers when incorporated into ointment and oleogel formulations. Finally, they evaluated the antioxidant potency of methanolic extracts of human skin after applying different semi-solid formulations containing phenolic compounds. The results indicated that the oleogel formulation, containing a complex of phenolic compounds, was the most effective antioxidant product for skin penetration.

Similarly, Rafique et al. [120] developed different cosmetic formulations (emulgel and emulsion) using grape seed extract (GSE) and assessed their efficacy and quality. The results revealed that GSE-based formulations effectively increased skin elasticity and hydration while reducing roughness, scaliness, and wrinkles. Although both formulations were effective in moisturizing and anti-aging the skin, the emulgel formulations demonstrated better results.

In conclusion, evaluating penetration is crucial when assessing the efficacy of cosmetics as the effectiveness of active ingredients depends on their ability to penetrate the skin and exert their effects. 

#### 3.1.5. Evaluation of the Stability of Cosmetic Formulation with Phenolic Compounds

According EU regulation [91,96], studying the stability of cosmetic formulations is crucial to ensure that the product remains safe, effective, and aesthetically pleasing throughout its shelf life. Cosmetic products are formulated with a variety of ingredients that may interact with each other, undergo chemical or physical changes, or degrade over time. These changes can result in reduced product effectiveness, altered sensory properties, and potential safety issues for consumers.

The stability of a cosmetic product is typically evaluated through a range of physical, chemical, and microbiological tests, including pH measurement, viscosity determination, color and odor evaluation, microbial contamination testing, and accelerated aging studies. These tests help to identify changes in the product’s appearance, texture, and chemical composition over time and assess the product’s potential for degradation or microbial growth.

In this sense, Rafique et al. [111] tested the stability of their formulation by observing the physical characteristics of the emulsion, such as liquefaction, coalescence, partial and complete separation of phases, pH measurements, color, and conductivity, for 12 weeks. The results showed that the formulation is stable at room temperature for 12 weeks. During this period, no changes were observed in the appearance of the product, only a small drop in the pH of the formulation, from 5.36 (0 h) to 4.75 (12 weeks). However, this pH value remained in the normal range of pH of human skin (4.5–6.0).

Wagemaker et al. [108] submitted their formulations to centrifugation, pH measurements, microscopy, visual observation, and viscosity to assess the preliminary stability of the product. Its formulation did not present phase separation or the formation of crystalline structures, which indicated that the formulation is stable to be applied on the skin and penetrate it. The pH values of the formulations remained between 5.07 and 6.53, and there was an increase in viscosity in the first 7 days, which was maintained after 45 days.

After developing several cosmetic formulations based on grape seed extracts, Rafique et al. [120] determined their stability. The formulations maintained their stability and viscosity for 12 weeks. The pH values of the formulations remained between 4.6 and 5.8, with no phase separation observed.

Aiming to study the stability of sunscreens with grape seed extracts, Yarovaya et al. [115] submitted the formulations to heat/cold cycles and then evaluated them. The results revealed that the formulations showed good stability after the test, the white color was maintained, as well as the light and smooth lotion-like texture.

Thus, stability is an important feature to ensure because this can also help to maintain the quality and efficacy of the product throughout its shelf life, ensuring that consumers receive the intended benefits of the phenolic compounds for their skin health and beauty.

### 3.2. Delivery Systems

One promising approach to harness the potential of phenolic compounds from grape stems involves incorporating them into advanced delivery systems. These delivery systems, which can include nanoparticles, liposomes, and other specialized carriers, offer unique advantages for enhancing the therapeutic efficacy of these compounds in skincare and cosmetic products [121,122,123,124].

By encapsulating phenolic compounds within these delivery systems, several benefits can be achieved. Firstly, the encapsulation protects the compounds from degradation and oxidation, ensuring their stability and preserving their bioactivity. This is particularly important for phenolic compounds, as they can be sensitive to environmental factors and lose their efficacy when exposed to light, air, or heat [123,124].

Moreover, these delivery systems provide controlled-release mechanisms, allowing for the gradual and sustained release of phenolic compounds over an extended period. This controlled release is crucial for achieving long-lasting effects on the skin. By releasing the compounds slowly, the skin can continuously benefit from their antioxidant, anti-inflammatory, and anti-aging properties, promoting skin health and rejuvenation.

Furthermore, the use of specialized carriers such as nanoparticles and liposomes enables targeted delivery of phenolic compounds to specific skin layers or cells. These carriers can penetrate the skin barrier and deliver the encapsulated compounds to the desired site of action, maximizing their therapeutic effects. For example, nanoparticles can be designed to target deep skin layers, while liposomes can deliver compounds to the surface layers of the skin. This targeted delivery enhances the efficiency and specificity of phenolic compound action, optimizing their benefits for different skin concerns [121,122,123,124].

Incorporating phenolic compounds into advanced delivery systems also offers opportunities for formulation versatility. These systems can be combined with various cosmetic formulations, including creams, serums, lotions, and masks, allowing for diverse product options that cater to different skin types and desired outcomes. This versatility expands the potential applications of phenolic compounds in the cosmetic industry and provides consumers with a wide range of effective skincare solutions [121,122].

In summary, by incorporating phenolic compounds from grape stems into advanced delivery systems, their therapeutic efficacy in skincare and cosmetic products can be significantly enhanced. These delivery systems provide protection, controlled release, and targeted delivery, allowing for optimal utilization of the bioactive properties of these compounds. By utilizing these innovative approaches, the cosmetic industry can develop products that offer long-lasting, targeted, and effective skincare benefits, meeting the growing demand for natural and sustainable beauty solutions [121,122,123,124].

## 4. Conclusions

The use of grape stem extracts in cosmetics holds great promise for unlocking a wide range of beauty benefits. These extracts are rich in phenolic compounds, which have been found to possess antioxidant, anti-inflammatory, antimicrobial, and anti-aging properties, among others. By leveraging these compounds, grape stem extracts have the potential to shield the skin from oxidative stress, reduce inflammation, and promote a more youthful appearance by enhancing skin elasticity and firmness.

While research on the application of grape stem extracts in cosmetics is still in its early stages, the results thus far are highly encouraging. Utilizing these extracts can offer a natural and sustainable alternative to conventional synthetic cosmetic ingredients. Additionally, incorporating grape stem extracts supports the concept of a circular economy by repurposing a by-product of the wine industry that would otherwise go to waste. As more research is conducted on the benefits of grape stem extracts, it is likely that their incorporation into cosmetics will become more prevalent. Cosmetic companies can harness the potential of grape stem extracts in their formulations, providing consumers with safe, effective, and sustainable products that promote healthy and beautiful skin. In recent times, there have been promising strategies to overcome challenges associated with delivering phenolic compounds to the skin. One approach involves incorporating these compounds into delivery systems such as nanoparticles, liposomes, and other specialized carriers. 

## Figures and Tables

**Figure 1 ijms-24-11751-f001:**
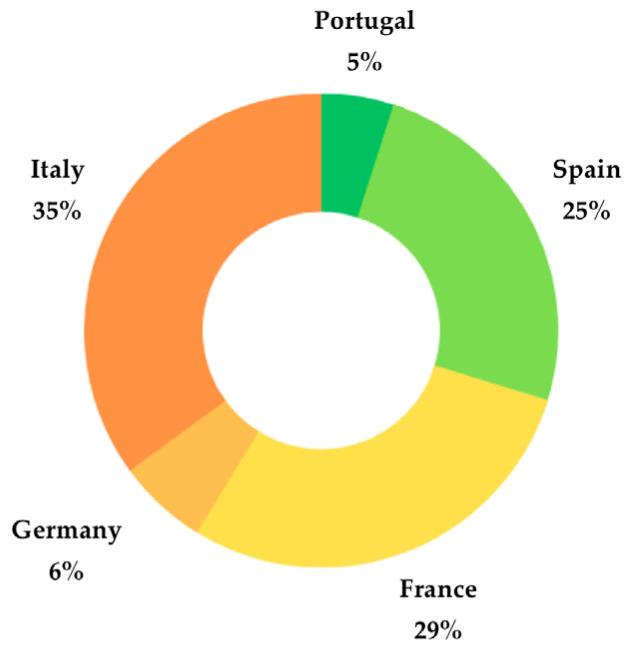
Percentual distribution of the main European wine producer countries.

**Figure 2 ijms-24-11751-f002:**
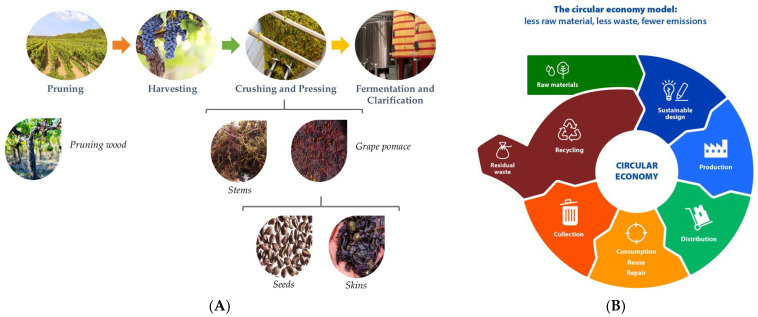
(**A**). Representation of the winemaking process and generated by-products. (**B**). The circular economy model according to European Parliament [18].

**Figure 3 ijms-24-11751-f003:**
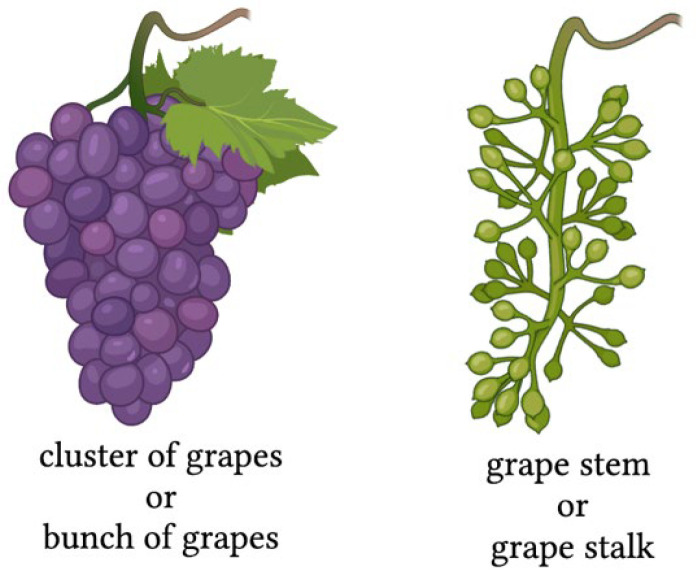
Schematic representation of the bunch and stem of the grape.

**Figure 4 ijms-24-11751-f004:**
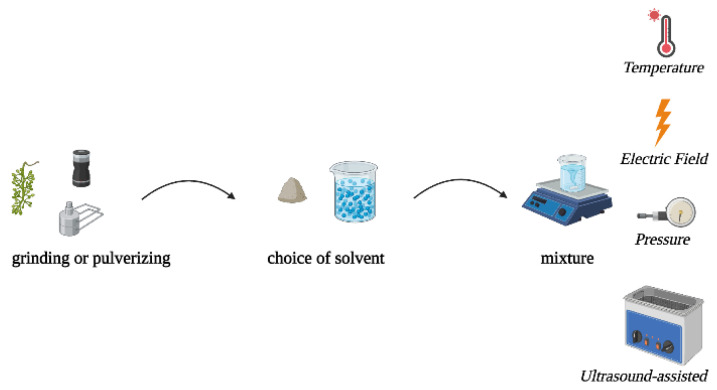
Schematic representation of the extraction procedure.

**Figure 5 ijms-24-11751-f005:**
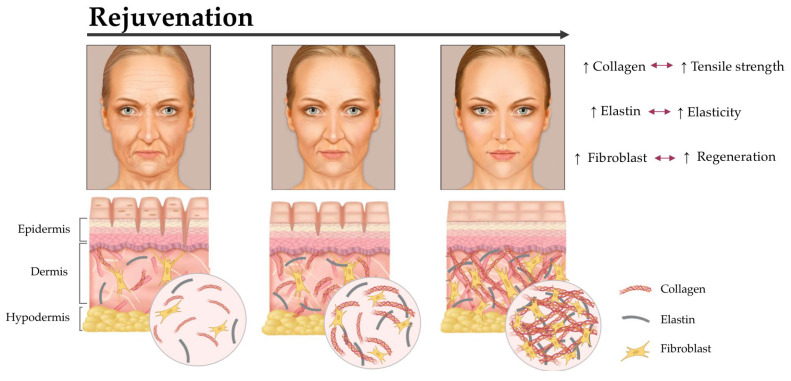
Schematic representation of the rejuvenation process. Adapted from www.freepik.com/vectors/skin-aging (accessed on 13 July 2023).

**Table 1 ijms-24-11751-t001:** Chemical structure of some examples of phenolic compounds.

Class of Phenolic Compounds	Examples
Flavan-3-ols	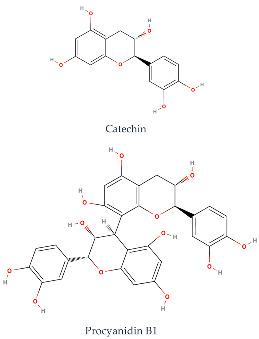
Flavonols	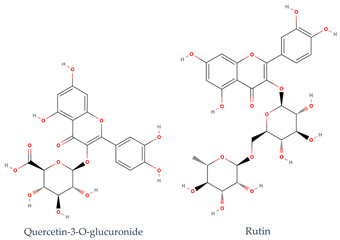
Hydroxybenzoic acids	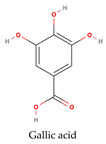
Hydroxycinnamic acids	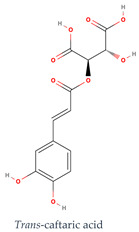
Stilbenes	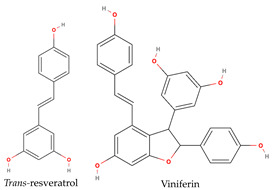

## Data Availability

Not applicable.

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
