# Peer review of "Revealing the Beauty Potential of Grape Stems: Harnessing Phenolic Compounds for Cosmetics"

_ijms, 2023, doi:10.3390/ijms241411751_

Round 1

Reviewer 1 Report

The subject of the manuscript is topical, very significant and interesting. The review article entitled: “ Revealing the Beauty Potential of Grape Stems: Harnessing Phenolic Compounds for Cosmetics “  is highly interesting and truly comprehensive. It extensively cites numerous authors and describes a multitude of fascinating studies and results. However, in my opinion, the review article requires restructuring and revision as the different points (sections) are not well-organized. Additionally, there are multiple spelling and style errors that need to be addressed. Furthermore, the literary sources should be arranged in journal format, both within the text and in the bibliography at the end.

Minor editing of English language required

Author Response

Dear Editor of International Journal of Molecular Sciences,

In reply to the review performed on the paper entitled “Revealing the Beauty Potential of Grape Stems: Harnessing Phenolic Compounds for Cosmetics”, we would like to acknowledge the valuable comments performed by the editor that kindly accepted to revise our manuscript. We would like to confirm that we have addressed all issues made by reviewer 1. We hope the answers below and modifications that have been done in the manuscript are clear and concise enough as required by the reviewer to enable the publication of the manuscript in International Journal of Molecular Sciences.

Answer to referee’s comments and queries

 Detailed responses to Reviewer 1

Reviewer´s comment: The subject of the manuscript is topical, very significant and interesting. The review article entitled: “ Revealing the Beauty Potential of Grape Stems: Harnessing Phenolic Compounds for Cosmetics “  is highly interesting and truly comprehensive. It extensively cites numerous authors and describes a multitude of fascinating studies and results. However, in my opinion, the review article requires restructuring and revision as the different points (sections) are not well-organized. Additionally, there are multiple spelling and style errors that need to be addressed. Furthermore, the literary sources should be arranged in journal format, both within the text and in the bibliography at the end.

Our reply: The authors would like to express their gratitude for all the comments provided by the reviewer. We would also like to inform that the article has been extensively rewritten in several sections to address the areas of improvement highlighted during the review process.

Sincerely,

Ana Isabel Ramos Novo Amorim de Barros

Reviewer 2 Report

The text below, contains comments on manuscript entitled “Revealing the Beauty Potential of Grape Stems: Harnessing Phenolic Compounds for Cosmetics”.

I think that the manuscript has a good idea to present the possibilities for utilization of grape stems and their incorporation in cosmetic products. The manuscript is well written on sufficiently good English. I am listing several comments that the authors might take into consideration if they find them useful.

To my opinion figures 2 and 3 can be presented in one figure as A and B. Both figures do not show any specific process, but general procedures winemaking processes and circular economy model.

Figure 4 and 5 can also be presented in one for the same reasons.

More chemical structures should be presented in Table 1. As you describe in the text there are much more structures than illustrated in table 1.

To my opinion if you just want to present the structure of the human skin layer, then I think you do not need this figure. But if want to present the difference between non-treated and treated skin with cosmetic product, illustrating the activated mechanisms that lead to skin improvement state, then I think this figure will be more complete.

If I have understood correctly the aim of the manuscript is focused on the utilization of grape stems in cosmetic products. Then I think that the section with the phenolic extraction is better to be more compact.

In section 3. Cosmetic Industry I think it will be good if you make a summary of cosmetic products based on grape or grape stems if available, e.g. the name of the product, producer, clinical trials, doses and duration of treatment, main effects, e.t.c. It also good if there is information which molecular mechanisms are activated in the skin by the cosmetic products.

Minor editing of English language required

Author Response

Dear Editor of International Journal of Molecular Sciences,

In reply to the review performed on the paper entitled “Revealing the Beauty Potential of Grape Stems: Harnessing Phenolic Compounds for Cosmetics”, we would like to acknowledge the valuable comments performed by the editor that kindly accepted to revise our manuscript. We would like to confirm that we have addressed most issues and answered the questions made by reviewer 2. We hope the answers below and modifications that have been done in the manuscript are clear and concise enough as required by the reviewer to enable the publication of the manuscript in International Journal of Molecular Sciences.

Answer to referee’s comments and queries

 Detailed responses to Reviewer 2

Reviewer´s comment: The text below, contains comments on manuscript entitled “Revealing the Beauty Potential of Grape Stems: Harnessing Phenolic Compounds for Cosmetics”.

I think that the manuscript has a good idea to present the possibilities for utilization of grape stems and their incorporation in cosmetic products. The manuscript is well written on sufficiently good English. I am listing several comments that the authors might take into consideration if they find them useful.

To my opinion figures 2 and 3 can be presented in one figure as A and B. Both figures do not show any specific process, but general procedures winemaking processes and circular economy model.

Our reply: The authors are grateful for the reviewer's comments and inform that the figures 2 and 3 are now Figure 2A and Figure 2B, and the text has also been changed according to the suggestion.

Reviewer´s comment: Figure 4 and 5 can also be presented in one for the same reasons.

Our reply: The authors are grateful for the reviewer's comments; however, we think that the figure about the extraction methodology must be in that topic. We hope you can understand our opinion about this point.

 Reviewer´s comment: More chemical structures should be presented in Table 1. As you describe in the text there are much more structures than illustrated in table 1.

Our reply: The authors appreciate the reviewer's comment; however, we have chosen to select the most representative classes of compounds and present the structure of one compound from each class. Introducing additional structures would make the table excessively exhaustive, potentially diverting the focus from the objective of the article.

Reviewer´s comment: To my opinion if you just want to present the structure of the human skin layer, then I think you do not need this figure. But if want to present the difference between non-treated and treated skin with cosmetic product, illustrating the activated mechanisms that lead to skin improvement state, then I think this figure will be more complete.

Our reply: Thank you for the comment. We fully agree with the comment made, and we would like to inform you that we have replaced the previous figure with a new one that clearly illustrates the information explained in the manuscript.

Reviewer´s comment: If I have understood correctly the aim of the manuscript is focused on the utilization of grape stems in cosmetic products. Then I think that the section with the phenolic extraction is better to be more compact.

Our reply: The authors appreciate the relevance of the comment, as this particular section was indeed excessively detailed. Following the suggestion, we have rewritten this section in a more concise manner.

Reviewer´s comment: In section 3. Cosmetic Industry I think it will be good if you make a summary of cosmetic products based on grape or grape stems if available, e.g. the name of the product, producer, clinical trials, doses and duration of treatment, main effects, e.t.c. It also good if there is information which molecular mechanisms are activated in the skin by the cosmetic products.

Our reply:

Thank you for your comment. Currently, there are no cosmetic products commercially available that are specifically formulated with grape stems. However, at the company Mesosystem, we have developed creams with varying concentrations of bioactive compounds derived from grape stems. We are conducting ongoing research and hope to publish a paper on this topic in the near future, providing more detailed information about our formulations and their effects. Regarding your suggestion for a summary of cosmetic products based on grape or grape stems, including the product name, producer, clinical trials, doses, duration of treatment, main effects, and molecular mechanisms activated in the skin, it is a valuable suggestion. While we don't have this summary available at the moment, we acknowledge the importance of providing comprehensive information about existing products and their mechanisms of action. We will consider developing such a summary in the future to enhance the understanding and awareness of cosmetic products utilizing grape or grape stem extracts.

Sincerely,

Ana Isabel Ramos Novo Amorim de Barros

Reviewer 3 Report

The review is generally good and interesting. Some revisions must be needed.

Line numbers should be added.

The introduction is too short. It should provide what novel information the author provided and how (systemically) the review obtain that information.

Figure 2 does not clearly show where the pomace comes from.

Table 1 needs references.

Delivery systems should be included in an individual section (half to one page at least) for a more comprehensive review.

Generally good

Author Response

Dear Editor of International Journal of Molecular Sciences,

In reply to the review performed on the paper entitled “Revealing the Beauty Potential of Grape Stems: Harnessing Phenolic Compounds for Cosmetics”, we would like to acknowledge the valuable comments performed by the editor that kindly accepted to revise our manuscript. We would like to confirm that we have addressed most issues and answered the questions made by reviewer 3. We hope the answers below and modifications that have been done in the manuscript are clear and concise enough as required by the reviewer to enable the publication of the manuscript in International Journal of Molecular Sciences.

Answer to referee’s comments and queries

 Detailed responses to Reviewer 3

Reviewer´s comment: The review is generally good and interesting. Some revisions must be needed.

Line numbers should be added.

Our reply: Thank you for your comment. We apologize for the oversight regarding the absence of lines. The lines have been added to the article as requested. We appreciate you bringing this to our attention and ensuring that the information is presented in a clearer and more organized manner.

Reviewer´s comment: The introduction is too short. It should provide what novel information the author provided and how (systemically) the review obtain that information.

Our reply: The authors are grateful for the reviewer's comments. We fully agree with this pertinent comment, and we would like to inform you that the introduction has been completely revised to better focus on the topic. Additionally, important information that was missing has been added to the article. We appreciate your feedback and suggestions, as they have helped us improve the overall quality and content of the paper.

Reviewer´s comment: Figure 2 does not clearly show where the pomace comes from.

Our reply: The authors appreciate the comment and would like to inform you that Figure 2 has been revised to clearly indicate the origin of each by-product. This modification aims to improve the clarity and understanding of the information presented in the figure. Thank you for bringing this to our attention, as it has allowed us to enhance the visual representation of the data in the article.

Reviewer´s comment: Delivery systems should be included in an individual section (half to one page at least) for a more comprehensive review.

Our reply: The authors are grateful for the reviewer's comments. Including a dedicated section on delivery systems is indeed a valuable suggestion for providing a more comprehensive review of the topic. Delivery systems play a crucial role in optimizing the effectiveness and efficiency of various active compounds, including those used in cosmetics, pharmaceuticals, and other applications.

By dedicating a section to delivery systems, it allows for a focused exploration of the different types of delivery systems available, their characteristics, and their impact on the delivery and release of bioactive compounds. This section can delve into the principles behind various delivery systems, such as nanoparticles, liposomes, microspheres, and emulsions, among others.

An individual section about this topic has been included in the manuscript.

Sincerely,

Ana Isabel Ramos Novo Amorim de Barros

Round 2

Reviewer 2 Report

The text below, contains comments on manuscript entitled “Revealing the Beauty Potential of Grape Stems: Harnessing Phenolic Compounds for Cosmetics” after first revision

To my opinion the manuscript is significantly approved. Although the authors have not considered all reviewer comments I fully understand their reasons and I’ll be looking forward to see their innovative results focused on grape stems cosmetic products.